# Label-Covariate Shift Chain: Unsupervised Domain Adaptation for Factorizable Joint Shift

## Abstract

The effectiveness of modern machine learning models for various tasks is fundamentally dependent on the presumption that training and test data are independent and identically distributed ($i.i.d.$). However, in some real-world scenarios, $i.i.d.$ is a luxury, i.e., distribution shifts often exist between training and test data. Factorizable joint shift is a new type of distribution shift, and unlike marginal shift (e.g., label shift or covariate shift) with strong assumptions, it imposes fewer constraints and provides broader applicability. However, unsupervised domain adaptation under factorizable joint shift is an unresolved and understudied problem. Previous methods easily collapse to trivial solutions, require the subjective selection of fixed constants, and fail to ensure the solution's existence and uniqueness when the number of categories exceeds two. To address this problem, we propose a principled method to find a non-trivial solution in a tractable manner. We first re-represent factorizable joint shift as a *Label-Covariate Shift Chain*, where label shift occurs first and then covariate shift occurs, which makes factorizable joint shift more tractable. Then, *Covariate Shift Minimization Principle* is introduced on the *Label-Covariate Shift Chain* to obtain a non-trivial solution. Furthermore, we propose a method to generate real-world factorizable joint shift datasets using *Label-Covariate Shift Chain*, and these datasets can serve as benchmarks to evaluate the effectiveness of generalization methods. Finally, the effectiveness of the proposed method is verified using real-world data for both accuracy improvement and confidence calibration tasks. We believe our exploration of factorizable joint shift will help modern machine learning models handle a wider variety of complex data scenarios, advancing the broader application of AI.

## 1 Introduction

Modern machine learning models, especially deep learning, have achieved tremendous success on various tasks (LeCun et al., 2015; Jiang et al., 2023). However, this success depends heavily on the fact that training data (or source domain) and test data (or target domain) are independent and identically distributed ($i.i.d.$) (Shao et al., 2024). In some real-world scenarios—such as medical diagnosis across different populations, image recognition under varying lighting conditions, or disease prediction during an epidemic—the $i.i.d.$ assumption does not hold, i.e., there is a distribution shift between the target domain and the source domain. This distribution shift will cause the trained model to experience catastrophic performance degradation on the target domain (Liang et al., 2025). Therefore, it is necessary to specialize methods to improve the model's generalization under distribution shift.

Without any assumption on the distribution shift, it's impossible to estimate how well the model would perform on the unlabeled new data (Chen et al., 2022; Tasche, 2022). Therefore, previous work mainly makes assumptions from two directions: 1) **Label shift**, where label distribution changes but the feature distribution under the label does not change (Lipton et al., 2018; Azizzadenesheli et al., 2019; Zhang et al., 2013; Guo et al., 2020; Tian et al., 2023; Wen et al., 2024); 2) **Covariate shift**, where the feature distribution changes but the label distribution under the feature condition does not change (Kimura & Hino, 2024; Sugiyama et al., 2007; Segovia-Martín et al., 2023; Cortes et al., 2010; Yamada et al., 2013; Rhodes et al., 2020; Fang et al., 2023; 2020). Al-

though label shift and covariate shift usefully characterize specific data distribution shifts, their underlying assumptions are overly restrictive. For example, they are inadequate in real-world scenarios where both label and covariate distributions change. He et al. (2022) propose a distribution shift assumption closer to the naive joint shift (i.e., no any assumption on joint distribution), named **Factorizable Joint Shift**, in which both the label distribution and the covariate distribution change mutually independently. Compared to label shift or covariate shift, factorizable joint shift imposes fewer constraints, covers more shift scenarios (including label and covariate shift), and has wider applicability. It offers a more flexible framework that better reflects some real-world data shift scenarios, e.g., medical diagnoses across diverse populations (covariate shift) amid incidence rate changes (label shift) during an epidemic.

However, how to generalize the model's performance under factorizable joint shift is an open problem, especially in the unsupervised domain adaptation scenario where the target domain's labels are not available. He et al. (2022) propose Joint Importance Aligning to address this problem, which determines the joint distribution's density ratio by solving a optimization with two additional deep learning models. However, He et al. (2022); Tasche (2023; 2022) indicated that the proposed method easily collapses to a trivial solution in the unsupervised domain adaptation if no additional assumptions are made, which limits its practicality. Tasche (2022) proposed an alternative method to Joint Importance Alignment from the perspective of measure theory. However, this method requires users to subjectively select fixed constants to solve the equation, and when the number of categories is greater than two, the existence and uniqueness of the solution become very complicated.

Therefore, a natural and necessary question is studied: How to find a non-trivial solution in a tractable manner for unsupervised domain adaptation under factorizable joint shift? To address this, we first re-represent factorizable joint shift as a *Label-Covariate Shift Chain*, where label shift occurs first and then covariate shift occurs. This representation's benefit is that it makes factorizable joint shift tractable and has the potential to leverage well-studied solutions for label shift and covariate shift to address factorizable joint shift. Then, an additional prior is introduced on the *Label-Covariate Shift Chain*: *Covariate Shift Minimization Principle*, which is used to determine non-trivial solutions. In addition, a real-world factorizable joint shift datasets generation method is proposed by using the *Label-Covariate Shift Chain*, and the generated datasets can be used to evaluate the effectiveness of generalization methods.

Our contributions can be summarized as follows:

- We prove that factorizable joint shift can be represented as a *Label-Covariate Shift Chain*, where label shift occurs first and then covariate shift occurs. This representation's benefit is that it makes factorizable joint shift tractable and has the potential to leverage well-studied solutions for label shift and covariate shift to address factorizable joint shift.

- *Covariate Shift Minimization Principle* is proposed in combination with *Label-Covariate Shift Chain* to determine non-trivial solutions for unsupervised domain adaptation under factorizable joint shifts.

- A real-world factorizable joint shift datasets generation method is proposed by using the *Label-Covariate Shift Chain*. The generated datasets can be used as benchmarks to evaluate the effectiveness of generalization methods.

## 2 BACKGROUND AND RELATED WORK

Consider a $K$-class classification problem where $X \in \mathcal{X}$ denotes the covariate variable and $Y \in \mathcal{Y}$ denotes the label variable, with $\mathcal{X} \subset \mathbb{R}^d$ and $\mathcal{Y} = \{1, 2, \ldots, K\}$. Let $p_s(\cdot)$ and $p_t(\cdot)$ denote the probability density (for continuous variables, e.g., $X$ and $X|Y$) or probability measure (for discrete variables, e.g., $Y$ and $Y|X$) on the source domain and target domain, respectively. Let $D_s = \{x_i, y_i\}_{1 \leq i \leq N_s}$ and $D_t = \{x_j\}_{1 \leq j \leq N_t}$ represent the source domain data and the target domain data, respectively, where $x_i$ (or $x_j$) represents the observed value of $X$, $y_i$ represents the observed value of $Y$, $N_s$ represents the sample size of the source domain, and $N_t$ represents the sample size of the target domain. Note that the target domain data has no labels, so the method studied in this paper is an unsupervised domain adaptation method.

## 2.1 Distribution Shift

Distribution shift refers to the situation where the joint distribution $p_t(X, Y)$ on the target domain differs from the joint distribution $p_s(X, Y)$ on the source domain. Since $p_s(X, Y) = p_s(X) \cdot p_s(Y|X) = p_s(Y) \cdot p_s(X|Y)$, the joint distribution $p_s(X, Y)$ will change if any of the components—$p_s(X)$, $p_s(Y|X)$, $p_s(Y)$, or $p_s(X|Y)$—undergo a change. Generally, when the underlying relationships between covariates and labels change (i.e., $p_s(Y|X)$ or $p_s(X|Y)$ changes), generalizing the trained model becomes more complex or even impossible (Chen et al., 2022; Tasche, 2022). Therefore, more attention is paid to marginal distribution shifts, i.e., label shift and covariate shift, as shown in Definition 1 and Definition 2 below.

**Definition 1. (Label Shift)** *Label shift occurs if the following two conditions are satisfied:* $p_s(Y) \neq p_t(Y)$ *and* $p_s(X|Y) = p_t(X|Y)$.

**Definition 2. (Covariate Shift)** *Covariate shift occurs if the following two conditions are satisfied:* $p_s(X) \neq p_t(X)$ *and* $p_s(Y|X) = p_t(Y|X)$.

**Definition 3. (Factorizable Joint Shift)** *Factorizable joint shift occurs if the following condition are satisfied:* $\frac{p_t(X,Y)}{p_s(X,Y)} = u(X) \cdot v(Y)$, *where* $u(\cdot)$ *and* $v(\cdot)$ *are functions on* $X$ *and* $Y$.

Although label shift and covariate shift effectively describe specific types of data distribution shifts, they fall short in scenarios where both the label and covariate distributions change. Therefore, He et al. (2022) propose Factorizable Joint Shift, a more relaxed joint shift assumption, as shown in Definition 3. It assumes that the joint density ratio can factorize the covariates and labels. Obviously, label shift and covariate shift are special factorizable joint shifts, corresponding to $u(X) \equiv 1$ and $v(Y) \equiv 1$ respectively. Therefore, factorizable joint shift covers more shift scenarios than label shift and covariate shift, and it has wider applicability.

## 2.2 Unsupervised Domain Adaptation

In the real world, the target domain's labels are usually unavailable. Therefore, how to generalize the model's performance without utilizing the target domain label information is crucial. This technique is called unsupervised domain adaptation. Specifically, we need design methods to estimate $p_t(Y|X)$ using $D_s = \{x_i, y_i\}_{1 \leq i \leq N_s}$ and $D_t = \{x_j\}_{1 \leq j \leq N_t}$, where $D_t$ does not include labels.

For factorizable joint shift, He et al. (2022) propose an unsupervised domain adaptation method named Joint Importance Aligning, which uses two deep learning models to learn $u(X)$ and $v(Y)$. Specifically, it estimates $u(X)$ and $v(Y)$ by minimizing the following formula:

$$\min_{\theta_u, \theta_v} \mathop{\mathbb{E}}_{p_s(X)} \log\left(1 + U(X; \theta_u)\tilde{V}(X; \theta_v)\right) + \mathop{\mathbb{E}}_{p_t(X)} \log\left(1 + 1/\left(U(X; \theta_u)\tilde{V}(X; \theta_v)\right)\right), \quad (1)$$

where $\tilde{V}(X; \theta_v) = \mathbb{E}_{Y \sim p_s(Y|X)} V(Y; \theta_v)$, $U(X; \theta_u)$ and $V(Y; \theta_v)$ represent two deep learning models, and $\theta_u$ and $\theta_v$ are the parameters of the deep learning models. Eq. 1 hopes that $U(X; \theta_u) \to u(X)$ and $V(Y; \theta_v) \to v(Y)$, and then generalize the model using importance weighting (Kimura & Hino, 2024). Optimizing Eq. 1 is equivalent to $p_t(X) = \sum_Y p_s(X, Y)u(X)v(Y)$ (He et al., 2022). However, the solution of $v(Y)$ in this equation is not unique and easily collapses to the trivial solution $v(Y) \equiv 1$ (i.e., $u(X) = p_t(X)/p_s(X)$). Therefore, Tasche (2022) proposed an alternative method to Joint Importance Alignment based on measure theory. However, this method requires the subjective selection of fixed constants, and ensuring the solution's existence and uniqueness becomes substantially more complex when the number of categories exceeds two. Therefore, it is necessary to study a tractable method to find a non-trivial solution for unsupervised domain adaptation under factorizable joint shifts.

## 3 Method

This section presents a tractable method to obtain a non-trivial solution in unsupervised domain adaptation under factorizable joint shift. Section 3.1 first proves that factorizable joint shift can be represented as a *Label-Covariate Shift Chain*, where label shift occurs first and then covariate shift occurs, which makes factorizable joint shift more tractable. Then, section 3.2 introduces *Covariate Shift Minimization Principle* on the *Label-Covariate Shift Chain* to obtain a non-trivial solution. Section 3.3 describes the empirical computation method for obtaining a non-trivial solution.

### 3.1 LABEL-COVARIATE SHIFT CHAIN

**Theorem 1. (Label-Covariate Shift Chain Theorem)** *If Factorizable Joint Shift occurs, then there exists a distribution $p_m(X, Y)$ such that:*

$$\frac{p_t(X, Y)}{p_s(X, Y)} = u(X) \cdot v(Y) = \frac{p_t(X)}{p_m(X)} \cdot \frac{p_m(Y)}{p_s(Y)}, \tag{2}$$

*and $p_m(X|Y) = p_s(X|Y)$ and $p_m(Y|X) = p_t(Y|X)$.*

*Proof.* First of all, construct a joint probability density $p_m(X, Y)$ such that: $p_m(X|Y) = p_s(X|Y)$, and $p_m(Y) = p_s(Y) \cdot v(Y)/C$, where $C$ is any constant greater than 0. Then, perform a covariate shift on $p_m(X, Y)$ to obtain a new distribution $p_n(X, Y)$, such that: $p_n(Y|X) = p_m(Y|X)$ and $p_n(X) = p_m(X) \cdot u(X) \cdot C$. Therefore:

$$\frac{p_n(X, Y)}{p_s(X, Y)} = \frac{p_n(X, Y)}{p_m(X, Y)} \cdot \frac{p_m(X, Y)}{p_s(X, Y)} = \frac{p_n(X)}{p_m(X)} \cdot \frac{p_m(Y)}{p_s(Y)} = u(X) \cdot C \cdot \frac{v(Y)}{C} = \frac{p_t(X, Y)}{p_s(X, Y)}. \tag{3}$$

Therefore $p_t(X, Y) = p_n(X, Y)$, i.e., Eq. 2 holds. □

**Remark of Theorem 1:** Theorem 1 tells us that Factorizable Joint Shift can be expressed as a *Label-Covariate Shift Chain*, i.e., label shift and covariate shift occur sequentially. Note that $p_m(X)$ and $p_m(Y)$ are unknown because $u(X)$ and $v(Y)$ are unknown. Even, $p_m(X)$ and $p_m(Y)$ are not unique because $C$ is uncertain. However, $u(X) \cdot v(Y)$ is unique, i.e., $\frac{p_t(X)}{p_m(X)} \cdot \frac{p_m(Y)}{p_s(Y)}$ is unique. Therefore, we can use Theorem 1 to construct a suitable $p_m(X, Y)$ to solve $u(X) \cdot v(Y)$ or $p_t(Y|X)$.

**Corollary 1.** *In supervised domain adaptation, from Theorem 1, it holds:*

$$p_t(X) = u(X) \sum_Y p_s(Y, X) v(Y), \tag{4a}$$

$$p_t(Y) = v(Y) \int_X p_s(X, Y) u(X) dX. \tag{4b}$$

*In addition, solving Eq. 4 (including Eq. 4a and Eq. 4b) can obtain the unique value of $u(X) \cdot v(Y)$. The proof is given in Appendix A.*

**Remark of Corollary 1:** Corollary 1 states that solving Eq. 4 yields the joint density ratio $u(X)v(Y)$. Careful observation reveals that: Eq. 4a corresponds to the unsupervised objective of Joint Importance Alignment (He et al., 2022) (see Section 2.2). In practice, Eq. 4b can not be used in unsupervised domain adaptation since $p_t(Y)$ is unavailable. Therefore, the value of $u(X)v(Y)$ cannot be determined using only Eq. 4a. This is exactly the reason why the unsupervised objective of Joint Importance Alignment collapses to a trivial solution (He et al., 2022). Therefore, the next question is how to add appropriate additional constraints based on Eq. 4a to find a non-trivial solution.

### 3.2 COVARIATE SHIFT MINIMIZATION PRINCIPLE

To obtain $p_t(Y|X)$ in an unsupervised situation, additional computable constraints must be added to Eq. 4a. If a model is trained with a distribution close to the target domain distribution, the model's generalization performance in the target domain will naturally be good. Based on this idea, we propose to construct a $p_m(X)$ that is closest to $p_t(X)$, and then use $p_m(X, Y)$ to train the model. Since there is a covariate shift between $p_m(X, Y)$ and $p_t(X, Y)$, $p_m(Y|X) = p_t(Y|X)$. Therefore, the trained model approaches $p_t(Y|X)$ when it approaches $p_m(Y|X)$. Moreover, when the supports of $p_m(X)$ and $p_t(X)$ differ substantially, methods that can learn new supports in an unsupervised manner can be leveraged to further fine-tune the trained classifier, e.g., pseudo-label training (Hu et al., 2021), and consistency regularization (Fan et al., 2023).

Formally, $p_m(X)$ can be constructed to close to $p_t(X)$ by optimizing the following formula:

$$\min_{\theta_v} \mathbb{E}_X \left[ L\left( p_t(X), \sum_Y p_s(Y|X) p_s(X) V(Y; \theta_v) \right) \right] + \lambda \left( 1 - \sum_Y p_s(Y) V(Y; \theta_v) \right)^2, \tag{5}$$

where $L(\cdot)$ is a loss function, $\lambda$ is the Lagrange multiplier. The first term of Eq. 5 holds because:

$$p_m(X) = \sum_Y p_m(X|Y)p_m(Y) = \sum_Y p_s(X|Y)p_s(Y)v(Y) = \sum_Y p_s(Y|X)p_s(X)v(Y). \quad (6)$$

Eq. 6 includes the information of Eq. 4a. The second term of Eq. 5 is to ensure $\sum_{y=1}^{K} p_m(y) = \sum_{y=1}^{K} p_s(y)V(y;\theta) = 1$. To make the output of $p_m(y)$ between 0 and 1, it is recommended to let $V(y;\theta) = \mathrm{sigmoid}(logit)/p_s(y)$, where $\mathrm{sigmoid}(\cdot)$ represents sigmoid activation function and $logit$ represents the output of the nueral network. After determining $v(Y)$ by optimizing Eq. 5, we then use the resampling technique to perform label shift on the source domain dataset $D_s$ to obtain new dataset $D_m$ (its distribution follows $p_m(X,Y)$). Then, using $D_m$ trains a model to obtain the potentially most generalizable model. Finally, if the supports of $p_m(X)$ and $p_t(X)$ differ substantially, or to avoid wasting the unlabeled target domain data $D_t$, methods that can learn new supports in an unsupervised manner can be leveraged to further fine-tune the trained model.

### 3.3 Empirical Computation

To optimize Eq. 5, $p_t(X)$, $p_s(X)$, $p_s(Y|X)$, and $p_s(Y)$ need to be properly estimated. An empirical estimation scheme is given below.

Thanks to the development of normalizing flow models, probability density calculation of high-dimensional random variables becomes efficient and exact (Kobyzev et al., 2021; Papamakarios et al., 2021). Therefore, we can use normalizing flow models (Zhai et al., 2025) to calculate $p_s(X)$ and $p_t(X)$ in Eq. 5. Specifically, let $F(X)$ be the output of the normalizing flow model, $F(X) \sim \mathcal{N}(0, I)$ be the latent variable output by the normalizing flow model, where $I$ is the identity matrix and $\mathcal{N}(0, I)$ is a multivariate normal distribution. Therefore:

$$\begin{cases} p_s(X) = \dfrac{\left|\det(J_{F_s(X)})\right|}{\sqrt{(2\pi)^d}} e^{-\frac{1}{2}\|F_s(X)\|^2}, \\[3mm] p_t(X) = \dfrac{\left|\det(J_{F_t(X)})\right|}{\sqrt{(2\pi)^d}} e^{-\frac{1}{2}\|F_t(X)\|^2}, \end{cases} \quad (7)$$

where $F_s(X)$ represents the output of the normalizing flow model on source domain, $F_t(X)$ represents the output of the normalizing flow model on target domain, $d$ represents the dimension of $X$, and $\det(J_{F_s(X)})$ and $\det(J_{F_t(X)})$ represent determinants of Jacobian matrix. In practice, if the value of $d$ is too large, the values of $p_s(X)$ and $p_t(X)$ will be too small to be calculated. Fortunately, this situation can be avoided by setting a specific loss function. Specifically, let $L(a,b) = (\log(a) - \log(b))^2$, then Eq. 5 gets rid of the dependence on $d$ and becomes:

$$\min_{\theta_v} \mathbb{E}_X\left[\left(\log g(X) - \log \sum_Y p_s(Y\,|\,X)\,V(Y;\theta_v)\right)^2\right] + \lambda\left(1 - \sum_Y p_s(Y)\,V(Y;\theta_v)\right)^2, \quad (8)$$

where the expression of $g(X)$ is as follows:

$$g(X) = \frac{\left|\det(J_{F_t(X)})\right|}{\left|\det(J_{F_s(X)})\right|} \cdot e^{\frac{\|F_s(X)\|^2 - \|F_t(X)\|^2}{2}}. \quad (9)$$

$p_s(Y|X)$ can be naturally obtained from the classifier on the source domain. $p_s(Y)$ can be estimated unbiasedly through frequency estimating probability, i.e., $p_s(y) \approx N_s^{(y)}/N_s$, where $N_s^{(y)}$ represents the sample size of the $y$-th class in the source domain. Let $\theta_v^*$ be the optimal solution of Eq. 8, and $p_m(Y) = p_s(Y)V(Y;\theta_v^*)$. Then, the samples of each class in $D_s$ are resampled with probability $p_m(Y)$ to obtain $D_m$. The generalization process using *Covariate Shift Minimization Principle* is shown in Algorithm 1 of Appendix B. Specifically, after determining $p_m(Y)$ by Eq. 8, the source domain data is resampled to simulate label shift, and a classifier is trained on the resampled data. Then, to further improve generalization, unsupervised fine-tuning is performed on the target domain to learn new supports in an unsupervised manner.

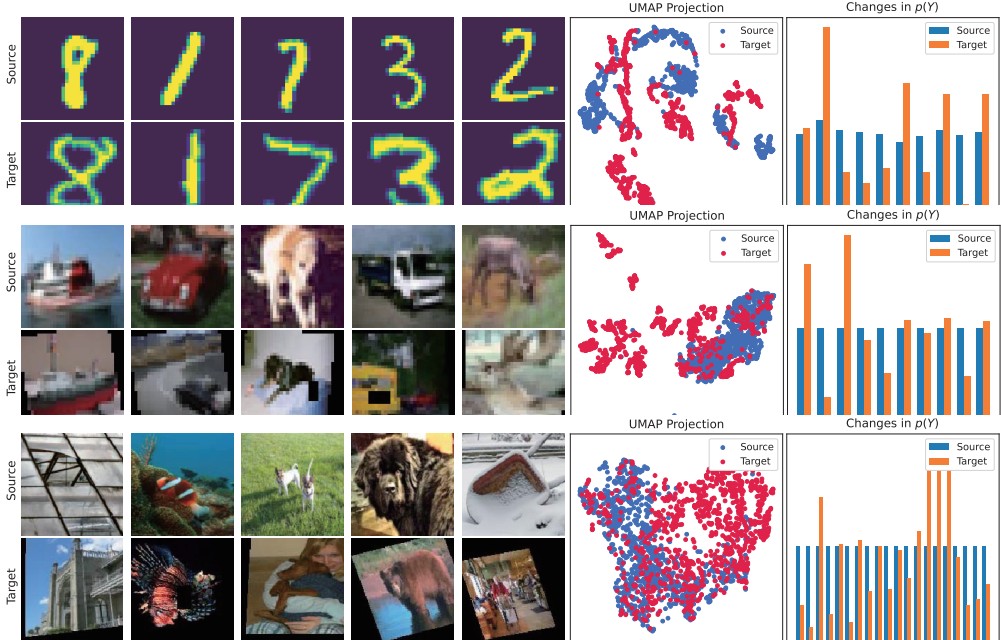

Figure 1: Visualization of generated factorizable joint shift dataset. UMAP Projection (Healy & McInnes, 2024) shows the covariate shift, and changes in $P(Y)$ shows the label shift. ImageNet-1K just shows 20 classes. The online implementation code is at: `https://github.com/Anonymous-user-code/LCSC/blob/main/GenerateFJSdata.ipynb`. Dirichlet distribution's concentration parameter is 2.

## 4   GENERATE FACTORIZABLE JOINT SHIFT DATASET

A key challenge in studying factorizable joint shifts is the lack of corresponding real-world datasets to validate existing methods. Therefore, it is necessary to propose a factorizable joint shift dataset generation method to advance the development of this field.

Another benefit of representing the factorizable joint shift as a *Label-Covariate Shift Chain* is that this representation can be used to generate benchmark datasets. These datasets, simulating real-world factorizable joint shift, allow for evaluating the effectiveness of generalization methods. Specifically, from Theorem 1, a factorizable joint shift dataset can be obtained by performing label shift and covariate shift on the real-world dataset in sequence. Algorithm 2 of Appendix B shows the process of generating a factorizable joint shift dataset. Specifically, the generation process involves two sequential steps: label shift followed by covariate shift. First, label shift is performed by resampling the source domain data according to a target label distribution $p_m(Y)$, which is typically initialized using a Dirichlet distribution to control the degree of imbalance. Then, covariate shift is introduced by resampling and followed by transforming the resampled data using predefined data augmentation techniques (e.g., rotation, cropping, brightness adjustment). Resampling is to change the sampling frequency of samples independently of the labels, and data transformation is to change the position of the support points.

## 5   RESULTS

### 5.1   GENERATE DATASETS

**Experimental Setup:** To demonstrate the universality of Algorithm 2, we generate Factorizable Joint Shift Dataset on three datasets with different sizes: 1) A grayscale digit recognition dataset **MNIST** (Lecun et al., 1998); 2) A colorful real-world image recognition dataset **CIFAR-10** (Krizhevsky, 2009); 3) A large-scale color real-world image recognition dataset **ImageNet-1K** (Deng et al., 2009). $p_m(Y)$ and $p_t(X)$ in Algorithm 2 are initialized by Dirichlet distribution sam-

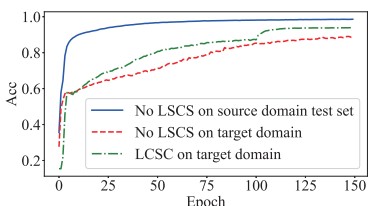

Figure 2: Visualization of the training process on MNIST. The curve with LCSC is significantly higher than the curve without LCSC on the target domain, indicating that LCSC improves the accuracy on the target domain. The results provide strong empirical evidence that LCSC performs domain adaptation on joint shift data. The online implementation code is available at: https://github.com/Anonymous-user-code/LCSC/blob/main/GeneralizationTrain.ipynb

Table 1: Classification accuracy on the generated factorizable joint shift dataset. Results (mean $\pm$ std) over 10 runs. No-FT represents no fine-tuning.

| Methods | MNIST | CIFAR-10 | ImageNet-1K |
|---|---|---|---|
| UnAdapt | $84.90_{\pm 0.86}$ | $64.32_{\pm 1.05}$ | $58.17_{\pm 0.92}$ |
| BBSE | $88.13_{\pm 0.65}$ | $66.55_{\pm 0.93}$ | $62.31_{\pm 1.01}$ |
| RLLS | $88.15_{\pm 0.67}$ | $66.56_{\pm 0.91}$ | $62.26_{\pm 0.97}$ |
| EM | $88.16_{\pm 0.70}$ | $66.62_{\pm 0.82}$ | $62.83_{\pm 1.10}$ |
| CPMCN | $88.37_{\pm 0.77}$ | $66.91_{\pm 0.81}$ | $63.24_{\pm 0.95}$ |
| LSC | $88.26_{\pm 0.69}$ | $66.43_{\pm 0.85}$ | $62.79_{\pm 0.88}$ |
| DANN | $89.85_{\pm 0.60}$ | $67.92_{\pm 0.77}$ | $64.12_{\pm 0.88}$ |
| TENT | $90.12_{\pm 0.58}$ | $68.21_{\pm 0.74}$ | $64.65_{\pm 0.86}$ |
| DIW | $90.54_{\pm 0.55}$ | $68.43_{\pm 0.73}$ | $64.97_{\pm 0.89}$ |
| DUA | $90.25_{\pm 0.49}$ | $68.75_{\pm 0.78}$ | $65.23_{\pm 0.95}$ |
| IndUDA | $91.37_{\pm 0.58}$ | $68.91_{\pm 0.74}$ | $64.99_{\pm 0.99}$ |
| GIW | $89.12_{\pm 0.61}$ | $67.05_{\pm 0.80}$ | $63.58_{\pm 0.93}$ |
| DW-GCS | $89.64_{\pm 0.50}$ | $67.38_{\pm 0.81}$ | $64.91_{\pm 0.92}$ |
| RSW | $90.85_{\pm 0.51}$ | $68.82_{\pm 0.75}$ | $65.10_{\pm 0.90}$ |
| JIA | $88.97_{\pm 0.51}$ | $68.35_{\pm 0.69}$ | $65.02_{\pm 0.80}$ |
| AJIA | $90.02_{\pm 0.57}$ | $68.22_{\pm 0.66}$ | $64.87_{\pm 0.92}$ |
| LCSC (No-FT) | $92.06_{\pm 0.47}$ | $68.79_{\pm 0.71}$ | $65.13_{\pm 0.84}$ |
| LCSC (Ours) | $\mathbf{94.66}_{\pm 0.44}$ | $\mathbf{71.86}_{\pm 0.76}$ | $\mathbf{68.53}_{\pm 0.80}$ |

pling. The degree of label shift can be controlled by adjusting the concentration parameter of the Dirichlet distribution, as detailed in Appendix C.2. $N_t$ is set to be as large as the test set's sample size. Algorithm 2's data transformation methods are presented in Appendix C.1.

**Experimental Results:** Fig. 1 shows three generated Factorizable Joint Shift Datasets. MNIST at the top, followed by CIFAR-10, with ImageNet-1K appearing at the bottom. On the far left, five source domain samples and five target domain samples are given, respectively. The Umap Projection (Healy & McInnes, 2024) of the source domain dataset and the target domain dataset are given in the middle of each row of Fig. 1. The target domain projection points diverge significantly from the source domain points, demonstrating a substantial covariate shift. The right side of Fig. 1 shows the label distribution of the source domain dataset and the target domain dataset. By comparison, significant label shifts exist. Since label shift and covariate shift are performed sequentially, factorizable joint shifts exist between the target domain and the source domain. To the best of our knowledge, this is the first factorizable joint shift dataset from the real world.

## 5.2 ACCURACY IMPROVEMENT

### 5.2.1 EXPERIMENTAL SETUP

To more comprehensively assess the effectiveness of the proposed method, the following methods are compared: 1) **UnAdapt:** models trained on source data without domain adaptation processing; 2) Five baseline label shift solutions: **BBSE** (Lipton et al., 2018), **RLLS** (Azizzadenesheli et al., 2019), **EM** (Alexandari et al., 2020), **CPMCN** (Wen et al., 2024), and **LSC** (Wei et al., 2024); 3) Eight baseline covariate shift solutions: **DANN** (Ganin et al., 2016), **TENT** (Wang et al., 2021), **DIW** (Fang et al., 2020), **DUA** (Mirza et al., 2022), **IndUDA** (He et al., 2023a), **GIW** (Fang et al., 2023), **DW-GCS** (Segovia-Martín et al., 2023), and **RSW** (He et al., 2023b); 4) Two baseline factorizable joint shift solutions: Joint Importance Aligning (**JIA**) (He et al., 2022) and an Alternative of Joint Importance Aligning (**AJIA**) (Tasche, 2022); 5) **LCSC**: the proposed Algorithm 1. Due to limited space, other settings are shown in Appendix C.1.2.

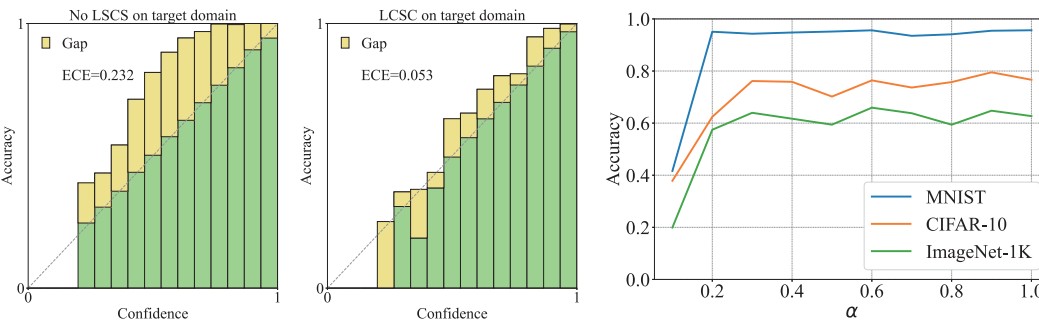

Figure 3: LCSC calibrates confidence on MNIST.  Figure 4: Selection experiments of $\alpha$.

### 5.2.2 Experimental Results

Fig. 2 shows the accuracy of the training process on the generated factorizable joint shift MNIST data, and see Appendix C.3 for other data. As training progresses, the classifier's accuracy on both the source domain test set and the target domain gradually increases. However, the accuracy on the target domain is significantly lower than that on the source domain test set, demonstrating that the factorizable joint shift leads to a decrease in the model's generalization performance. The accuracy curve of our generalization method LCSC in the target domain has been higher than that of the naive classifier after 20 epochs, indicating that LCSC can indeed generalize the model under factorizable joint shift. In addition, from the comparison of the accuracy curves before and after 100 epochs, our method can achieve a certain generalization effect regardless of whether unsupervised fine-tuning (i.e., line 23 in Algorithm 1) is used or not.

Table 1 compares classification accuracies of baseline methods in generated factorizable joint shift data. Furthermore, to observe the effect of LCSC under more realistic covariate shifts, we compare the accuracy improvement effect on the public domain shift datasets, see Appendix C.3. All methods all improve the classification accuracy on the target domain to a certain extent. Generally, methods that address covariate shift tend to have higher classification accuracy than those that address label shift. The accuracy improvement of JIA and AJIA (specialized in factorizable joint shift) is also limited because they are essentially importance weighting methods that cannot overcome the information bias caused by changes in support points. Similarly, without fine-tuning, LCSC yields only modest accuracy gains, since resampling alone cannot correct the inherent information bias introduced by shifts in the covariate support. As expected, LCSC with fine-tuning achieves the highest classification accuracy on the target domain because it simultaneously addresses both covariate shift and label shift, and can overcome the information bias caused by the change of support points.

### 5.3 Confidence Calibration

Due to space limitations, the experimental setup for confidence calibration is presented in Appendix C.4. Fig. 3 illustrates the effectiveness of confidence calibration on the MNIST dataset, and results for other datasets are provided in Appendix C.4. Compared with the reliability diagram obtained by the classifier without LCSC, the reliability diagram obtained by the classifier with LCSC is closer to the diagonal line, indicating the predicted confidence is more accurate. The classifier using LCSC achieves a much lower expected calibration error ($ECE$), showing its effectiveness in confidence calibration. To more comprehensively assess the calibration effectiveness of our method, some confidence calibration baseline methods are compared in Appendix C.4.

### 5.4 Ablation Experiment

**Impact of Lagrange Multiplier $\lambda$:** From Eq. 8, the value of $\lambda$ determines the importance of the second term (used to ensure $\sum_{y=1}^{K} p_m(y) = 1$). If $\lambda$ is too large, the model pays more attention to the constraint term and ignores the density ratio matching of the first term. If $\lambda$ is too small, the constraints may not be adequately satisfied. To select a suitable $\lambda$, we employed the gradual increase strategy. Specifically, let $\lambda = \alpha \cdot loss_1$, where $loss_1$ is the first loss in Eq. 8, and the value of $\alpha$ increases from 0.1 to 1. The selection experiment of $\alpha$ is shown in Fig. 4. When $\alpha$ is less than 0.3,

as $\alpha$ increases, the accuracy of the proposed method in the target domain is improved. However, when $\alpha$ is greater than 0.3, the accuracy of the proposed method in the target domain does not show an obvious increase or decrease. Therefore, we set $\alpha$ to 0.3 for all experiments in this paper.

Due to space limitations, we report other ablation experiments in the Appendix: **Selection of Unsupervised Fine-Tuning Methods** (Appendix C.5.1), **Impact of Density Estimation Effectiveness** (Appendix C.5.2), and **Practical Comparison of $p_m(Y)$ and $p_t(Y)$** (Appendix C.5.3).

## 6 DISCUSSION

**Reasonableness of Factorizable Joint Shift Assumption:** Real-world joint shifts often exhibit a factorizable structure: label distribution and covariate distribution change simultaneously but independently of each other. For example, the proportion of positive cases can surge during an epidemic (label shift) while CT images differ across hospitals due to variations in scanners and imaging protocols (covariate shift); urban scenes contain more cars and buses than rural areas (label shift), while lighting and background styles vary with region and time of day (covariate shift). In addition, since the study of joint shift is too difficult, most of the current work focuses on marginal assumptions (label shift or covariate shift). Thus, compared to the assumption limitations of these works, we have relaxed the assumptions and taken an important step forward.

**Differences from Conventional Unsupervised Domain Adaptation:** Conventional unsupervised domain adaptation (or out-of-distribution adaptation) primarily addresses covariate shift (Liu et al., 2022), typically through techniques such as feature alignment (Chen et al., 2019; Shi et al., 2024), ensemble learning (Zhou et al., 2021; Yang et al., 2024), or normalization statistics adjustment (Li et al., 2018; Zhang et al., 2024). In contrast, our method is designed to handle the more complex scenario involving both covariate shift and label shift, thereby offering broader applicability across diverse domain adaptation settings.

**Computational Overhead**: The computational overhead of LCSC is reported in Appendix D. Although LCSC introduces additional computation during training—primarily due to the two normalizing flow models—inference remains unaffected since only a classifier is used at test time. Typically, computing resources are less constrained during training. In our MNIST and CIFAR-10 experiments, a single NVIDIA GeForce RTX 3090 GPU (24 GB VRAM) was sufficient. For ImageNet-1K, using a single NVIDIA A100 GPU (80 GB VRAM) also yielded competitive results. Therefore, the computational overhead of density estimation in LCSC does not pose a significant barrier to its widespread adoption.

**Potential Impact, Limitations and Future Work:** We go beyond covariate shift or label shift to consider the more challenging factorizable joint shift and its generalization methods. We also generate factorizable joint shift datasets that can help subsequent studies evaluate the effectiveness of generalization methods. We believe this work has the potential to inspire a wealth of follow-up research, ultimately enhancing decision-making in real-world applications, particularly for cross-populations and safety-critical scenarios. However, our study also has several limitations: 1) Algorithm 1 needs to use the normalizing flow models to estimate the probability density of high-dimensional random variables, which is a computationally expensive operation in the training phase. In the future, Algorithm 1 will benefit from more efficient probability density estimation methods of high-dimensional random variables; 2) We did not consider the more naive joint shift. Although highly challenging or even intractable, further exploration of this concept remains intriguing.

## 7 CONCLUSION

This paper proposes a tractable generalization method for factorizable joint shift. Firstly, we re-represent factorizable joint shift as a *Label-Covariate Shift Chain*, where label shift occurs first and then covariate shift occurs. This representation makes factorizable joint shift more tractable. Then, an additional prior is introduced on the *Label-Covariate Shift Chain*: *Covariate Shift Minimization Principle*, which is used to determine non-trivial solutions. In addition, a real-world factorizable joint shift datasets generation method is proposed by using the *Label-Covariate Shift Chain*, and the generated datasets can be used to evaluate the effectiveness of generalization methods. Finally, experimental results verify the proposed method's generalization effectiveness.

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

## APPENDIX

## A   PROOF OF COROLLARY 1

*Proof.* From Theorem 1, the following holds:

$$
\begin{aligned}
p_t(X) &= \sum_Y p_t(X,Y) = \sum_Y p_t(Y|X)p_t(X) \\
&\underline{\text{Covariate shift}} \sum_Y p_m(Y|X)p_m(X)u(X)\cdot C = C\cdot u(X)\sum_Y p_m(X|Y)p_m(Y) \\
&\underline{\text{Label shift}}\, C\cdot u(X)\sum_Y p_s(X|Y)p_s(Y)\frac{v(Y)}{C} = u(X)\sum_Y p_s(Y,X)v(Y).
\end{aligned}
\tag{10}
$$

Similarly, the following holds:

$$p_t(Y) = \int_X p_t(X,Y)dX = \int_X p_t(Y|X)P_t(X)dX$$

$$\underline{\text{Covariate shift}} \int_X p_m(Y|X)p_m(X)C \cdot u(X)dX = \int_X p_m(X|Y)p_m(Y)C \cdot u(X)dX \quad (11)$$

$$\underline{\text{Label shift}} \int_X p_s(X|Y)p_s(Y)\frac{v(Y)}{C}C \cdot u(X)dX = v(Y)\int_X p_s(X,Y)u(X)dX.$$

Therefore, Eq. 4 is proved.

Below, prove the uniqueness of $u(X) \cdot v(Y)$. Proof by contradiction is used. Assume there are two solutions: $(u(X), v(Y))$ and $(u'(X), v'(Y))$. By Eq. 4, it holds:

$$p_t(X) = u(X)\sum_Y p_s(Y,X)v(Y) = u'(X)\sum_Y p_s(Y,X)v'(Y),$$

$$p_t(Y) = v(Y)\int_X p_s(X,Y)u(X)dX = v'(Y)\int_X p_s(X,Y)u'(X)dX. \quad (12)$$

Let $r(X) = u'(X)/u(X)$, $s(Y) = v'(Y)/v(Y)$, $a_X(Y) = \frac{p_s(Y,X)v(Y)}{\sum_{Y'} p_s(Y',X)v(Y')}$, and $b_Y(X) = \frac{p_s(X,Y)u(X)}{\int_X p_s(X,Y)u(X)dX}$. Therefore:

$$1 = r(X)\frac{\sum_Y p_s(Y,X)s(Y)v(Y)}{\sum_Y p_s(Y,X)v(Y)} = r(X)E_{Y \sim a_X}[s(Y)],$$

$$1 = s(Y)\frac{\int_X p_s(X,Y)r(X)u(X)dX}{\int_X p_s(X,Y)u(X)dX} = s(Y)E_{X \sim b_Y}[r(X)]. \quad (13)$$

Therefore, get a fixed point:

$$r(X) = \frac{1}{E_{Y \sim a_X}\left[\frac{1}{E_{X' \sim b_Y}[r(X')]}\right]}. \quad (14)$$

Let $r(X)$ reach its minimum value $m = \inf_X r(X)$ at $X^*$. Due to Eq. 14:

$$m = \frac{1}{E_{Y \sim a_{X^*}}\left[\frac{1}{m}\right]} = r(X^*) = \frac{1}{E_{Y \sim a_{X^*}}\left[\frac{1}{E_{X' \sim b_Y}[r(X')]}\right]}. \quad (15)$$

Therefore:

$$E_{Y \sim a_{X^*}}\left[\frac{1}{m}\right] = E_{Y \sim a_{X^*}}\left[\frac{1}{E_{X' \sim b_Y}[r(X')]}\right]. \quad (16)$$

Therefore:

$$E_{Y \sim a_{X^*}}\left[\frac{1}{m} - \frac{1}{E_{X' \sim b_Y}[r(X')]}\right] = 0. \quad (17)$$

Due to $m \le r(X')$ for all $X'$, and then $\frac{1}{m} - \frac{1}{E_{X' \sim b_Y}[r(X')]} \ge 0$. Therefore $m = E_{X' \sim b_Y}[r(X')]$. Similarly, due to $m \le r(X')$ for all $X'$, $r(X') = m$ for all $b_Y(X') \ne 0$. Therefore, $r(X)$ is a constant function in $p_s(X,Y) \ne 0$. Let $r(X) = C$, then:

$$s(Y) = \frac{1}{E_{X \sim b_Y}[r(X)]} = \frac{1}{C}. \quad (18)$$

Therefore:

$$u'(X)v'(Y) = (u(X)r(X))(v(Y)s(Y)) = C \cdot u(X)\frac{1}{C}v(Y) = u(X)v(Y). \quad (19)$$

Therefore, the uniqueness of $u(X) \cdot v(Y)$ is proved. $\square$

---

**Algorithm 1** Generalization by Covariate Shift Minimization Principle.

---

1: **Initialize:**
2:   $D_s = \{x_i, y_i\}_{1 \le i \le N_s}$; $D_t = \{x_j\}_{1 \le j \le N_t}$; $N_m$;
3:   $D_s^{(k)} = \{x | (x, y) \in D_s$ and y = k$\}$; $D_m = \{\}$;
4:   A normalizing flow model $F_s(X)$;
5:   A normalizing flow model $F_t(X)$;
6:   A classifier $f_1(X)$; A classifier $f_2(X)$;
7:   A parametric model $V(Y; \theta_v)$.
8: **Training:**
9:   Using $D_s^x = \{x | (x, y) \in D_s\}$ to train $F_s(X)$.
10:   Using $D_t$ to train $F_t(X)$.
11:   Using $D_s$ to train $f_1(X)$.
12:   Get $\theta_v^*$ by optimizing Eq. 8.
13: **Resampling:**
14:   Let $h_k = \sum_{y=1}^{k} p_s(y) V(y; \theta_v^*)$, where $1 \le k \le K$.
15:   $i = 0$.
16:   **While** $i \le N_m$:
17:     Uniformly sample a number $h$ in $[0, 1]$.
18:     If $h_{k-1} < h \le h_k$:
19:       Uniformly select a sample from $D_s^{(k)}$ to $D_m$.
20:       $i = i + 1$.
21: **Generalization Training:**
22:   Using $D_m$ to train $f_2(X)$.
23:   Unsupervised fine-tuning $f_2(X)$ on $D_t$.
24: **Return** $f_2(X)$.

---

## B  PSEUDO-CODE

**Generalization by Covariate Shift Minimization Principle:** Algorithm 1 implements the Covariate Shift Minimization Principle on the proposed Label–Covariate Shift Chain to obtain a non-trivial estimate of $p_t(Y|X)$ using only labeled source data $D_s$ and unlabeled target data $D_t$. Its main steps are as follows: (i) estimate source and target covariate densities using normalizing flows and train a source classifier; (ii) optimize Eq. 8 to learn $V(Y; \theta_v)$ and derive the adjusted label prior $p_m(Y) = p_s(Y) V(Y; \theta_v^*)$; (iii) resample the source dataset according to $p_m(Y)$ to simulate label shift, train a new classifier on the resampled data, and optionally fine-tune it on the unlabeled target domain. Unsupervised fine-tuning in line 23 of Algorithm 1 performs further generalization to learn new supports in an unsupervised manner.

**Generate Factorizable Joint Shift Dataset:** Algorithm 2 constructs a target domain exhibiting factorizable joint shift by sequentially applying label shift and covariate shift to a source dataset. First, a target label distribution $p_m(Y)$ is sampled (e.g., from a Dirichlet prior) to control the degree of imbalance, and source samples are resampled according to $p_m(Y)$ to form an intermediate dataset. Next, covariate shift is introduced by resampling and applying predefined transformations (e.g., rotation, cropping, brightness, or contrast adjustment) to alter the support of $X$ while preserving semantics. The resulting dataset $D_t$ thus differs from the source in both label and covariate distributions, providing a realistic benchmark for evaluating domain adaptation under factorizable joint shift. Note that to ensure the shift simulation rationality, $\sum_{y=1}^{K} p_m(y) = 1$ and $\sum_{j=1}^{N_t} p_t(x_j) \le 1$ must be satisfied when initializing $p_m(Y)$ and $p_t(X)$. In addition, both resampling and data transformation are used in covariate shifts. Resampling is to change the sampling frequency of samples independently of the labels, and data transformation is to change the position of the support points.

---

**Algorithm 2** Generate Factorizable Joint Shift Dataset.

---

1: **Initialize:**
2: $\quad D_s = \{x_i, y_i\}_{1 \leq i \leq N_s}; D_t = \{\};$
3: $\quad D_s^{(k)} = \{(x,y)|(x,y) \in D_s \text{ and } y = k\};$
4: $\quad N_t; D_m = \{\}; p_t(X); p_m(Y);$
5: **Label Shift:**
6: $\quad$ Let $h_k = \sum_{y=1}^{k} p_m(y)$, where $1 \leq k \leq K$.
7: $\quad i = 0.$
8: $\quad$ **While** $i \leq N_t$:
9: $\quad\quad$ Generate a uniform random number $h$ in $[0, 1]$.
10: $\quad\quad$ If $h_{k-1} < h \leq h_k$:
11: $\quad\quad\quad$ Uniformly sample $(x, y)$ from $D_s^{(k)}$ to $D_m$.
12: $\quad\quad\quad i = i + 1.$
13: **Covariate Shift:**
14: $\quad$ Let $h_J = \sum_{j=1}^{J} p_t(x_j)$, where $1 \leq J \leq N_t$.
15: $\quad i = 0.$
16: $\quad$ **While** $i \leq N_t$:
17: $\quad\quad$ Uniformly sample a number $h$ in $[0, h_{N_t}]$.
18: $\quad\quad$ If $h_{J-1} < h \leq h_J$:
19: $\quad\quad\quad$ Transform $x_J$ to obtain $x'_J$.
20: $\quad\quad\quad$ Add $(x'_J, y_J)$ to $D_t$.
21: $\quad\quad\quad i = i + 1.$
22: **Return** $D_t$.

---

## C RESULTS

### C.1 EXPERIMENTAL SETTINGS

All experiment was conducted on Intel® Core$^{TM}$ I7-10700 CPU with 3.70GHz and 125.5GB memory, 10 NVIDIA GeForce RTX 3090 graphics cards (each with 24GB of video memory), Ubuntu 20.04.3 LTS, Python 3.11.11, and Torch 2.3.0+cu121.

#### C.1.1 EXPERIMENTAL SETUP OF GENERATING DATASETS

Algorithm 2's data transformation methods applied to three datasets are presented in Table 2, which controls the degree of shift in the covariate support points. To ensure the semantics of the data remain unchanged, the rotation angles for MNIST and CIFAR-10 are $(-10°, +10°)$, and the rotation angles for ImageNet-1K are $(-15°, +15°)$. Regarding random shifting, all three datasets are shifted horizontally or vertically no more than 10% of the image size. Regarding random cropping, the image cropping sizes in the three datasets are all 87.5% of the source image size, and will be resized to the original size after cropping. Regarding brightness adjustment, the brightness adjustment range does not exceed 10% of the image pixel range. Specifically, $I_{out} = I_{in} + \Delta b$ for $\Delta b \in (-0.1 * r, +0.1 * r)$, where $I_{in}$ represents the input image, $I_{out}$ represents the output image, and $r$ represents the pixel range. Similarly, the contrast adjustment degree does not exceed 10%, i.e., $I_{out} = c \cdot (I_{in} - r_{mid}) + r_{mid}$ for $c \in [0.9, 1.1]$, where $r_{mid}$ represents the mid value of the pixel range. Regarding random scaling, the scaling range is set to $(0.85, 1.15)$. Regarding random erasing, the probability of erasing is set to 0.5. To ensure the semantics of the data remain unchanged, range of proportion of erased area against input image is $(0.01, 0.05)$. Note that the data transformation parameters above are user-specific settings that can be made based on the desired covariate shift requirements.

#### C.1.2 EXPERIMENTAL SETUP OF ACCURACY IMPROVEMENT

$N_m$ of Algorithm 1 is set to be as large as the train set's sample size. Depending on the size of the dataset, appropriate and popular classifiers are used for the corresponding dataset: LeNet-5 for MNIST, ResNet-56 for CIFAR-10, and ResNet-152 for ImageNet-1K. $f_1(X)$ and $f_2(X)$ in Algorithm 1 use classifiers with the same structure. The learning rate of LeNet-5 is 0.01. The learning rate of ResNet-56 and ResNet-152 are 0.1 for the first 50 epochs, 0.01 for 50 to 100 epochs,

Table 2: Data transformation methods on Algorithm 2. Note that data transformation is only used to change the position of covariate support, and only combined with resampling can a complete covariate shift be achieved (see line 17 of Algorithm 2).

| Datasets | Transformation Methods |
|----------|------------------------|
| MNIST | Random rotation ($\leq 10°$); Random shifting; Random cropping; Brightness adjustment; Contrast adjustment. |
| CIFAR-10 | Random cropping; Random rotation ($\leq 10°$); Random shifting; Random scaling; Random Erasing |
| ImageNet-1K | Random cropping; Random rotation ($\leq 15°$); Random shifting; Random scaling; Random Erasing |

and 0.001 after 100 epochs. The normalizing flow models' hyperparameter settings are shown in Table 3. $V(Y; \theta_v)$ is a simple three-layer feedforward neural network with the number of neurons in the hidden layer being twice the number of categories, trained with the Adam optimizer with a learning rate of 0.001. All classifiers are trained for 150 epochs, all normalizing flow models are trained for 100 epochs, and $V(Y; \theta_v)$ is trained for 100 epochs. In Algorithm 1's generalization train, unsupervised fine-tuning is performed using naive pseudo-label training. Specifically, the predicted confidences are sorted from largest to smallest, and then the top 75% of samples are used for pseudo-label training (see section C.5). In Algorithm 1's generalization train, the first 100 epochs are using $D_m$ to train $f_2(X)$, and the next 50 epochs are unsupervised fine-tuning $f_2(X)$ with pseudo labels.

Table 3: Hyperparameter settings for TarFlow (Zhai et al., 2025). P represents the patch size, Ch represents the model channel size, T represents the number of autoregressive flow blocks, K represents the number of attention layers in each flow, and $p_\epsilon$ represents the noise distribution.

| Datasets | P-Ch-T-K-$p_\epsilon$ | Optimizer | Learning rate |
|----------|----------------------|-----------|---------------|
| MNIST | 2-128-4-4-$\mathcal{N}(0, 0.1)$ | Adam | 0.002 |
| CIFAR-10 | 2-256-4-4-$\mathcal{N}(0, 0.05)$ | Adam | 0.0002 |
| ImageNet-1K | 4-768-8-8-$\mathcal{N}(0, 0.15)$ | Adam | 0.0001 |

## C.2 GENERATE DATASETS

$p_m(Y)$ in Algorithm 2 are initialized by Dirichlet distribution sampling. The degree of label shift can be controlled by adjusting the concentration parameter of the Dirichlet distribution, as shown in Fig. 5. The smaller the concentration parameter, the greater the imbalance of the sampled data; the larger the concentration parameter, the smaller the imbalance of the sampled data. This is because a smaller concentration parameter increases the probability that the Dirichlet distribution places samples near the edge of the simplex, while a larger concentration parameter causes the distribution to concentrate samples nearer the center of the simplex. Therefore, when the sample sizes of classes in the source domain are almost balanced, the degree of label shift can be controlled by adjusting the concentration parameter of the Dirichlet distribution.

## C.3 ACCURACY IMPROVEMENT

Fig. 6 shows the accuracy of the training process on CIFAR-10 and ImageNet-1K. As training progresses, the classifier's accuracy on both the source domain test set and the target domain gradually increases. However, the accuracy on the target domain is significantly lower than that on the source domain test set, demonstrating that the factorizable joint shift leads to a decrease in the model's generalization performance. Whether on CIFAR-10 or ImageNet-1K, the accuracy curve of our generalization method LCSC in the target domain has been higher than that of the naive classifier,

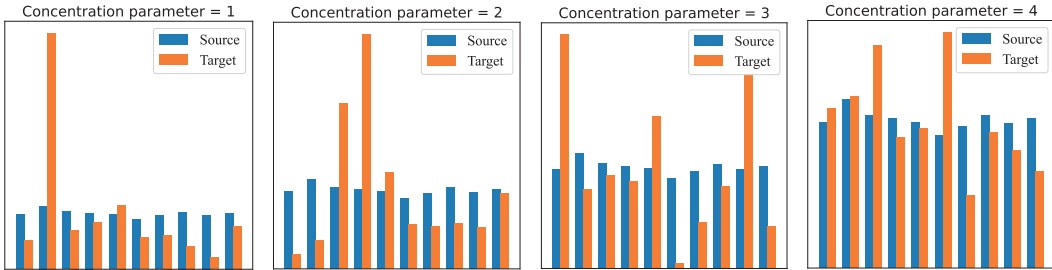

Figure 5: The degree of label shift can be controlled by adjusting Dirichlet distribution's concentration parameter in MNIST.

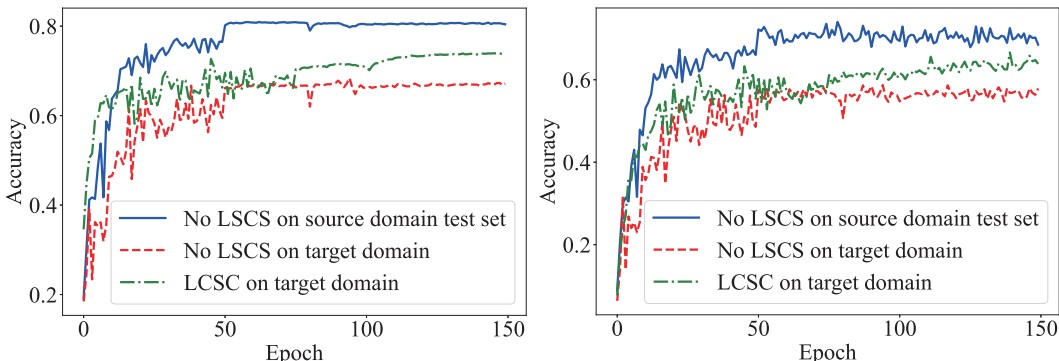

Figure 6: Visualization of the training processes. Results for CIFAR-10 (left) and ImageNet-1K (right).

indicating that LCSC can indeed generalize the model under factorizable joint shift. In addition, from the comparison of the accuracy curves before and after 100 epochs, our method can achieve a certain generalization effect regardless of whether unsupervised fine-tuning (i.e., line 23 in Algorithm 1) is used or not.

**Results on the Public Domain Shift Datasets:** To observe the effect of LCSC under more realistic covariate shifts, we compare the accuracy improvement effect on the public domain shift dataset. We considered the following three public domain shift datasets: 1) **NICO** (He et al., 2021); 2) **Office-Home** (Venkateswara et al., 2017); 3) **iWildCam** (Beery et al., 2021). In these three datasets, the first half of the domain index is used as the source domain, and the second half of the domain index is used as the target domain. To simulate the joint shift, the source domain is sampled relatively balanced, and the target domain is sampled into a simplex (the concentration parameter of the Dirichlet distribution is 2). Table 4 shows the results comparison on these three datasets. Methods designed for label shift correction (BBSE, RLLS, EM, CPMCN, LSC) provide moderate improvements over UnAdapt, while covariate shift adaptation methods (e.g., DANN, TENT, DIW, DUA, IndUDA) achieve higher gains, reflecting their ability to handle feature distribution changes. Specialized approaches for factorizable joint shift (JIA, AJIA) offer limited additional benefit compared to covariate shift methods. Finally, the proposed LCSC method shows the best performance: without fine-tuning (No-FT), it already matches or surpasses the strongest baselines, and with unsupervised fine-tuning (Ours), it achieves the highest accuracy on all datasets, demonstrating its effectiveness in addressing both label and covariate shifts simultaneously.

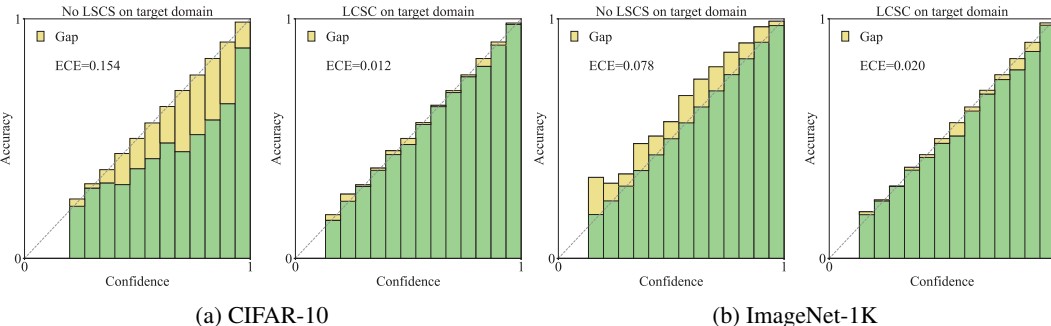

Figure 7: LCSC helps calibrate confidence.

Table 4: Classification accuracy on the generated factorizable joint shift dataset. Results (mean $\pm$ std) over 10 runs. No-FT represents no fine-tuning.

| Methods | NICO | Office-Home | iWildCam |
|---|---|---|---|
| UnAdapt | $54.60_{\pm 0.82}$ | $50.30_{\pm 0.93}$ | $60.30_{\pm 0.88}$ |
| BBSE | $58.90_{\pm 0.79}$ | $55.70_{\pm 0.90}$ | $63.70_{\pm 0.86}$ |
| RLLS | $59.00_{\pm 0.78}$ | $55.90_{\pm 0.89}$ | $63.60_{\pm 0.85}$ |
| EM | $59.20_{\pm 0.77}$ | $56.10_{\pm 0.88}$ | $63.90_{\pm 0.84}$ |
| CPMCN | $59.60_{\pm 0.75}$ | $56.50_{\pm 0.87}$ | $64.30_{\pm 0.84}$ |
| LSC | $59.10_{\pm 0.76}$ | $55.80_{\pm 0.88}$ | $63.80_{\pm 0.85}$ |
| DANN | $60.80_{\pm 0.74}$ | $57.80_{\pm 0.85}$ | $65.10_{\pm 0.82}$ |
| TENT | $61.30_{\pm 0.72}$ | $58.40_{\pm 0.84}$ | $65.60_{\pm 0.81}$ |
| DIW | $61.70_{\pm 0.71}$ | $58.80_{\pm 0.83}$ | $65.90_{\pm 0.81}$ |
| DUA | $62.10_{\pm 0.72}$ | $59.20_{\pm 0.83}$ | $66.20_{\pm 0.80}$ |
| IndUDA | $62.50_{\pm 0.70}$ | $59.60_{\pm 0.82}$ | $66.40_{\pm 0.80}$ |
| GIW | $61.00_{\pm 0.73}$ | $58.10_{\pm 0.84}$ | $65.30_{\pm 0.82}$ |
| DW-GCS | $61.40_{\pm 0.72}$ | $58.60_{\pm 0.83}$ | $65.70_{\pm 0.81}$ |
| RSW | $62.80_{\pm 0.71}$ | $59.80_{\pm 0.82}$ | $66.60_{\pm 0.80}$ |
| JIA | $61.50_{\pm 0.70}$ | $58.70_{\pm 0.82}$ | $65.80_{\pm 0.79}$ |
| AJIA | $61.20_{\pm 0.70}$ | $58.50_{\pm 0.82}$ | $65.70_{\pm 0.79}$ |
| LCSC (No-FT) | $62.90_{\pm 0.69}$ | $60.00_{\pm 0.80}$ | $66.70_{\pm 0.79}$ |
| LCSC (Ours) | $\mathbf{64.80}_{\pm 0.66}$ | $\mathbf{61.70}_{\pm 0.79}$ | $\mathbf{67.90}_{\pm 0.78}$ |

## C.4 CONFIDENCE CALIBRATION

**Experimental Setup:** Algorithm 1's generalization on the confidence calibration task is also veri-fied. To measure the accurateness of predicted confidence score, two of the most popular calibration metrics are adopted: reliability diagram (Dimitriadis et al., 2021), expected calibration error ($ECE$) (Guo et al., 2017), debiased calibration error ($ECE_{debiased}$) (Kumar et al., 2019), and calibration error using Kolmogorov-Smirnov test ($KS\text{-}error$) (Gupta et al., 2021). The bin number of confi-dence binning is set to the popular 15 (Dong et al., 2025b;a) when calculating reliability plots and $ECE$. Since the training process is shared with Section 5.2, its hyperparameters are identical to those in Section 5.2.

**Reliability Diagram:** Fig. 7 shows the effect of confidence calibration on the CIFAR-10 and ImageNet-1K. From the reliability diagram, the confidence scores on CIFAR-10 show overconfi-dence (the average accuracy is below the diagonal line), and the confidence scores on ImageNet-1K show underconfidence (the average accuracy is above the diagonal line). Whether on CIFAR-10 or ImageNet-1K, compared with the reliability diagram obtained by the classifier without LCSC, the reliability diagram obtained by the classifier with LCSC is closer to the diagonal line, indicating the predicted confidence is more accurate. In addition, the $ECE$ obtained by the classifier using LCSC is also significantly smaller, indicating that LCSC can indeed help calibrate confidence.

**Baseline Methods:** To more comprehensively assess the calibration effectiveness of the proposed method, the following calibration methods are compared: 1) **Uncal**: uncalibrated model trained on source data; 2) **TempScal**: calibration on source data using Temperature scaling (Guo et al., 2017); 3) Two confidence calibration methods under label shift: **LADE** (Hong et al., 2021) and **LaSCal** (Popordanoska et al., 2024); 4) Two confidence calibration methods under covariate shift: **TransCal** (Wang et al., 2020) and **PseudoCal** (Hu et al., 2024); 5) **LCSC**: the proposed Algorithm 1.

**Calibration metrics Comparison:** Across all three datasets (MNIST, CIFAR-10, ImageNet-1K) and all three calibration metrics($ECE$, $ECE_{debiased}$, and $KS\text{-}error$), LCSC consistently achieves the lowest error, indicating that its probability estimates align most closely with empirical accuracy and with the target confidence distribution, as shown in Table 5, Table 6 and Table 7. Notably, methods tailored to a single type of shift—LADE/LaSCal (label shift), and TransCal/PseudoCal (covariate shift)—provide meaningful but limited gains relative to the uncalibrated baseline, whereas LCSC yields uniformly larger reductions, especially on the more challenging CIFAR-10 and ImageNet-1K settings, where distribution shifts are stronger. The agreement of improvements across $ECE$ and $ECE_{debiased}$ suggests the effect is not an artifact of binning bias, and the concurrent decrease in $KS\text{-}error$ further confirms that LCSC improves the full calibration curve rather than only average bin deviations. We attribute these gains to LCSC's joint treatment of label and covariate shift on the Label–Covariate Shift Chain: aligning $p_m(X)$ toward $p_t(X)$ mitigates covariate mismatch while the learned prior $p_m(Y)$ prevents collapse to trivial importance weights, leading to better-calibrated posteriors under target distributional changes.

Table 5: $ECE$ (%) Comparison in Confidence Calibration Baseline Methods. Results (mean $\pm$ std) over 10 runs.

| Dataset | Uncal | TempScal | LADE | LaSCal | TransCal | PseudoCal | LCSC |
|---|---|---|---|---|---|---|---|
| MNIST | $12.47_{\pm 0.61}$ | $7.923_{\pm 0.44}$ | $5.368_{\pm 0.38}$ | $4.885_{\pm 0.35}$ | $4.116_{\pm 0.29}$ | $5.927_{\pm 0.41}$ | $\mathbf{2.354}_{\pm 0.21}$ |
| CIFAR-10 | $23.58_{\pm 0.73}$ | $15.84_{\pm 0.65}$ | $12.43_{\pm 0.57}$ | $11.38_{\pm 0.54}$ | $10.72_{\pm 0.49}$ | $12.91_{\pm 0.60}$ | $\mathbf{7.457}_{\pm 0.36}$ |
| ImageNet-1K | $31.92_{\pm 0.88}$ | $22.67_{\pm 0.79}$ | $18.34_{\pm 0.71}$ | $17.12_{\pm 0.69}$ | $16.48_{\pm 0.66}$ | $18.95_{\pm 0.75}$ | $\mathbf{12.03}_{\pm 0.52}$ |

Table 6: $ECE_{debiased}$ (%) Comparison in Confidence Calibration Baseline Methods. Results (mean $\pm$ std) over 10 runs.

| Dataset | Uncal | TempScal | LADE | LaSCal | TransCal | PseudoCal | LCSC |
|---|---|---|---|---|---|---|---|
| MNIST | $11.09_{\pm 0.54}$ | $6.87_{\pm 0.38}$ | $4.45_{\pm 0.32}$ | $3.97_{\pm 0.28}$ | $3.26_{\pm 0.23}$ | $5.00_{\pm 0.35}$ | $\mathbf{1.84}_{\pm 0.16}$ |
| CIFAR-10 | $21.49_{\pm 0.67}$ | $14.18_{\pm 0.58}$ | $10.64_{\pm 0.49}$ | $9.53_{\pm 0.45}$ | $8.83_{\pm 0.40}$ | $11.21_{\pm 0.52}$ | $\mathbf{5.98}_{\pm 0.29}$ |
| ImageNet-1K | $29.18_{\pm 0.80}$ | $20.47_{\pm 0.71}$ | $16.02_{\pm 0.62}$ | $14.67_{\pm 0.59}$ | $13.90_{\pm 0.56}$ | $16.91_{\pm 0.67}$ | $\mathbf{10.00}_{\pm 0.43}$ |

Table 7: $KS\text{-}error$ (%) Comparison in Confidence Calibration Baseline Methods. Results (mean $\pm$ std) over 10 runs.

| Dataset | Uncal | TempScal | LADE | LaSCal | TransCal | PseudoCal | LCSC |
|---|---|---|---|---|---|---|---|
| MNIST | $9.63_{\pm 0.47}$ | $5.85_{\pm 0.32}$ | $3.67_{\pm 0.26}$ | $3.19_{\pm 0.23}$ | $2.60_{\pm 0.18}$ | $4.20_{\pm 0.29}$ | $\mathbf{1.41}_{\pm 0.13}$ |
| CIFAR-10 | $18.91_{\pm 0.59}$ | $12.24_{\pm 0.50}$ | $8.94_{\pm 0.41}$ | $7.85_{\pm 0.37}$ | $7.08_{\pm 0.32}$ | $9.55_{\pm 0.44}$ | $\mathbf{4.65}_{\pm 0.22}$ |
| ImageNet-1K | $26.19_{\pm 0.72}$ | $17.89_{\pm 0.62}$ | $13.62_{\pm 0.53}$ | $12.18_{\pm 0.49}$ | $11.32_{\pm 0.45}$ | $14.56_{\pm 0.58}$ | $\mathbf{7.99}_{\pm 0.35}$ |

## C.5 ABLATION EXPERIMENTS

### C.5.1 SELECTION OF UNSUPERVISED FINE-TUNING METHODS

Algorithm 1's unsupervised fine-tuning is to use the available unlabeled target source data to learn new covariate supports in an unsupervised manner. We try two popular and simple methods: consistency regularization (Koh & Fernando, 2023) and pseudo-label training (Li et al., 2023). Additionally, we examined the impact of different pseudo-label sample ratios on generalization performance for pseudo-label training.

Table 8: Ablation experiments on unsupervised fine-tuning methods. CR represents consistency regularization.

| Methods | MNIST | CIFAR-10 | ImageNet-1K |
|---|---|---|---|
| UnAdapt | $84.90_{\pm 0.86}$ | $64.32_{\pm 1.05}$ | $58.17_{\pm 0.92}$ |
| CR | $83.25_{\pm 0.91}$ | $60.61_{\pm 0.99}$ | $57.88_{\pm 1.02}$ |
| Pseudo-label (25%) | $93.58_{\pm 0.79}$ | $70.53_{\pm 0.90}$ | $65.91_{\pm 0.91}$ |
| Pseudo-label (50%) | $93.21_{\pm 0.70}$ | $71.72_{\pm 0.80}$ | $67.66_{\pm 0.86}$ |
| Pseudo-label (75%) | $\mathbf{94.66}_{\pm 0.44}$ | $\mathbf{71.86}_{\pm 0.76}$ | $\mathbf{68.53}_{\pm 0.80}$ |
| Pseudo-label (100%) | $93.22_{\pm 0.73}$ | $70.29_{\pm 0.72}$ | $64.32_{\pm 0.83}$ |

Table 9: Impact of density estimation's effectiveness. The numbers in brackets are the classification accuracy on the source domain's test set for $p_s(Y|X)$ or bits per dimension (BPD) (Zhai et al., 2025) for $p_s(X)$ and $p_t(X)$.

| Method or Object | MNIST | CIFAR-10 | ImageNet-1K |
|---|---|---|---|
| UnAdapt | 84.90 | 64.32 | 58.17 |
| $p_s(Y|X)$ (75 epochs) | 93.51 (97.5) | 68.77 (84.3) | 64.41 (66.4) |
| $p_s(Y|X)$ (100 epochs) | 93.96 (98.0) | 70.29 (86.0) | 66.27 (69.0) |
| $p_s(Y|X)$ (125 epochs) | 94.14 (98.3) | 71.03 (88.7) | 67.92 (75.4) |
| $p_s(Y|X)$ (150 epochs) | $\mathbf{94.66}$ (98.5) | $\mathbf{71.86}$ (89.1) | $\mathbf{68.53}$ (76.3) |
| $p_s(X)$ (50 epochs) | 93.29 (4.90) | 70.35 (6.02) | 66.83 (6.19) |
| $p_s(X)$ (75 epochs) | 93.57 (4.03) | 70.84 (4.92) | 67.49 (5.07) |
| $p_s(X)$ (100 epochs) | $\mathbf{94.66}$ (3.51) | $\mathbf{71.86}$ (3.06) | $\mathbf{68.53}$ (3.95) |
| $p_t(X)$ (50 epochs) | 93.18 (3.75) | 70.30 (6.51) | 66.75 (6.10) |
| $p_t(X)$ (75 epochs) | 93.50 (2.98) | 70.89 (4.90) | 67.61 (5.20) |
| $p_t(X)$ (100 epochs) | $\mathbf{94.66}$ (2.04) | $\mathbf{71.86}$ (3.02) | $\mathbf{68.53}$ (3.83) |

Table 8 shows the ablation experiments on unsupervised fine-tuning methods. The main idea of consistency regularization is to perform weak augmentation and strong augmentation on the target domain image, respectively, and then make the prediction results of the two augmented images tend to be consistent. To our surprise, even when the strong augmentation is set to the same data transformation as in Table 2, the consistency regularization method does not generalize well to the target domain. This may be because when there is a distribution shift between the target domain and the source domain, even if the weakly augmented samples and the strongly augmented samples are consistent, it does not tell the model the class to which the target domain samples belong, so it cannot help improve the classification accuracy. This problem does not exist in pseudo-label training because it directly tells the model which class the samples in the target domain belong to. Therefore, pseudo-label training can improve the model's classification accuracy in the target domain.

In pseudo-label training, it is crucial to select which samples' prediction results to use as pseudo-labels. Therefore, in Table 8, experiments with different proportions of samples as pseudo labels were conducted (sorted by predicted confidence scores from largest to smallest), and it was ultimately found that selecting the top 75% of samples as pseudo labels yielded the best results.

### C.5.2 IMPACT OF DENSITY ESTIMATION EFFECTIVENESS

In Algorithm 1, $p_s(Y|X)$, $p_s(X)$, and $p_t(X)$ need to be estimated using network models. Therefore, their estimated effects may affect the performance of Algorithm 1. Here, we control their estimation effectiveness by early stopping the training, and then observe their impact on the performance of Algorithm 1. Table 9 shows their impact, where BPD is a popular evaluation metric for density estimation effect. It can be known that no matter for $p_s(Y|X)$, $p_s(X)$, or $p_t(X)$, the better the density estimation, the more the accuracy of Algorithm 1 in the target domain is improved. Moreover, even when the density estimator is not fully trained, Algorithm 1 shows significant accuracy improvements compared to other methods (see Table 1).

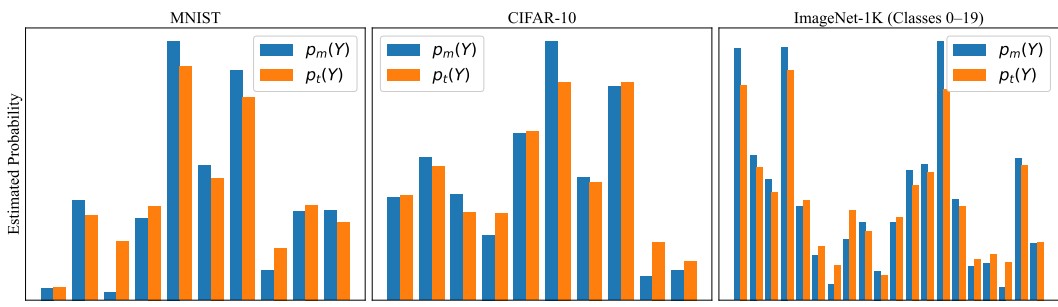

Figure 8: Practical Comparison of $\boldsymbol{p_m(Y)}$ and $\boldsymbol{p_t(Y)}$. Note that it is not necessary for $p_m(Y)$ to approximate $p_t(Y)$ in Algorithm 1.

### C.5.3 PRACTICAL COMPARISON OF $p_m(Y)$ AND $p_t(Y)$

Fig. 8 illustrates that the learned label distribution $p_m(Y)$ of the intermediate dataset, obtained via Eq. 8, often differs subtly from the label distribution $p_t(Y)$. This discrepancy is expected and does not undermine the effectiveness of LCSC, because the optimization objective focuses on aligning the marginal distribution $p_m(X)$ with $p_t(X)$ rather than matching label priors. In other words, $p_m(Y)$ serves as an instrumental prior to minimize covariate shift while preserving $p_m(X|Y) = p_s(X|Y)$, ensuring that the resulting model approximates $p_t(Y|X)$. Generally, the larger the covariate shift between the source and target domains, the less likely $p_m(Y)$ is to be close to $p_t(Y)$. If there is no covariate shift, and the joint shift degenerates into a label shift, then $p_m(Y)$ will approach $p_t(Y)$.

## D COMPUTATIONAL OVERHEAD

Our method introduces extra training-time cost mainly from two normalizing flow models for estimating $p_s(X)$ and $p_t(X)$. As quantified in Table 6, the flows dominate parameters and FLOPs across datasets (e.g., on ImageNet-1K, TarFlow has 460.8M params/931.45 GFLOPs versus the classifier's 2.98M/7.09 GFLOPs), whereas inference remains unaffected since only the classifier is used at test time. In practice, one NVIDIA RTX 3090 GPU (24GB VRAM) suffices for MNIST/CIFAR-10 and one NVIDIA A100 GPU (80GB VRAM) for ImageNet-1K, suggesting the training overhead is manageable; future work will explore more efficient high-dimensional density estimators to further reduce the cost.

Table 10: Computational Overhead Report. Param(M) represents the number of model parameters and the unit is mega. GFlops represent calculation amount.

| Datasets | Model | Param (M) | GFlops |
|---|---|---|---|
| MNIST | LeNet-5 | 0.0444 | 0.0006 |
| | TarFlow (2-128-4-4-$\mathcal{N}(0, 0.1)$) | 3.2794 | 0.6255 |
| CIFAR-10 | ResNet-56 | 0.8557 | 0.2547 |
| | TarFlow (2-256-4-4-$\mathcal{N}(0, 0.05)$) | 12.936 | 6.5017 |
| ImageNet-1K | ResNet-152 | 2.9797 | 7.0913 |
| | TarFlow (4-768-8-8-$\mathcal{N}(0, 0.15)$) | 460.80 | 931.45 |

## E DESCRIPTION OF LARGE LANGUAGE MODEL USAGE

We only used the large language model to polish the writing.

