# OpenReview forum: "Label-Covariate Shift Chain: Unsupervised Domain Adaptation for Factorizable Joint Shift"
_ICLR.cc/2026/Conference — ICLR 2026 Conference Withdrawn Submission_

### Official Review · Reviewer_vRN9 · 2025-10-29

**Soundness:** 2
**Presentation:** 2
**Contribution:** 2
**Rating:** 4
**Confidence:** 4

**Summary:**

This work considers a general shift correction framework for the unsupervised domain adaptation problem, named Factorizable Joint Shift, whose key idea is assuming that the joint shift can be decomposed as two independent terms that are dominated by variables $X$ and $Y$, respectively. Under such a framework, previous results only show a marginal alignment property in the unsupervised case, and the author shows that it would fall into a trivial solution. Motivated by this, this submission tries to explore some new properties under the Factorizable Joint Shift framework, and the derived results are useful. However, the major concern is the validity of theoretical results, where there are unclear points and non-negligible gap in the proof. Thus, it would be important to clarify the validity or the generality of the results.

**Strengths:**

+ The motivation of improving Factorizable Joint Shift is interesting and meaningful.

+ The organization is clear.

**Weaknesses:**

+ The validity of theoretical results is doubtful.

+ The significance of derived results would be reduced if the precondition is required to correct the proof.

**Questions:**

Q1. A primary concern is the validity of proof w.r.t. Theorem 1. In the proof of Theorem 1, there are two constructed distributions, i.e., $p_m(Y)$ and $p_n(X)$, that require a constant to normalize the mass such that they are distributions. However, there seems to be no justification for why the two distributions share the same constant $C$. That is to say, the constants in line 171 and line 173 formally should be $C_1$ and $C_2$, and it would be better to rigorously show that $C_1=C_2$.

Q2. Since the proof is problematic, the validity of Theorem 1 is doubtful (not a general conclusion at least). Specifically, there could be a potential contradiction considering the results and the proof. For example, considering $p_m(X)=\frac{p_t(X)}{u(X) \cdot C}$ (equation in Line 173) and $p_m(X)=\sum_Y p_m(X|Y) \cdot p_m(Y) = \sum_Y p_t(X|Y) \cdot \frac{p_s(Y) \cdot v(Y)}{C}$ (equations in Line 170-171), it directly obtains $u(X) \sum_Y p_t(X|Y) \cdot p_s(Y) \cdot v(Y) = p_t(X)$. Comparing the identity (4a) in Corollary 1, it shows that $p_s(X|Y)=p_t(X|Y)$, which implies the conditional distributions need to be invariant. Moreover, there could be more contradictions or potential requirements by considering the two equations mentioned in Q1. *Thus, I believe there should be some pre-conditions to ensure the validity of Theorem 1, and the result would not be as general as the current version*.

Q3. There seems to be a gap in Eq. (6). In the second equality, is the identity $p_m(Y)=p_s(Y) \cdot v(Y) \cdot C$ applied here? If $C=1$, it would be necessary to prove that $v(Y) p_s(Y)$ is still a distribution and $v(Y)$ should also satisfy the Factorizable Joint Shift condition, i.e., prove the existence of such $v(Y)$.

Q4. If the precondition is required, e.g., $p_s(X|Y)=p_t(X|Y)$, it would be crucial to discuss the significance of results under the precondition, as the key is the generality of derived results, which would be trivial under these preconditions.

---

### Official Review · Reviewer_ozX2 · 2025-10-31

**Soundness:** 3
**Presentation:** 2
**Contribution:** 2
**Rating:** 4
**Confidence:** 4

**Summary:**

The paper studies factorizable joint shift, where the label and covariate distributions change independently. Departing from prior work, authors prove that joint shift can be decomposed into a label-shift step followed by a covariate-shift  step. Building on this view, the authors propose minimizing the distributional discrepancy in $X$ between the target domain $T$ and a constructed intermediate domain $M$ to obtain more accurate predictions (i.e., a better approximation to $p_t(Y\mid X)$). They also outline a procedure for constructing datasets that exhibit joint shift. The experiments are promising and reasonably comprehensive, although evaluation on larger and more diverse datasets would further establish robustness and scalability.

**Strengths:**

* The paper offers a fresh perspective on factorizable joint shift.
* The proofs appear correct, and the proposed methods are compelling.
* The experimental study is substantial and well executed, the source code looks good.

**Weaknesses:**

- The paper lacks a concise summary of the overall model pipeline and the loss/objective functions; consider adding a schematic figure and a table listing each loss term with its definition and role.
- Line 172: C should be treated as a normalization constant, since pm(Y) is a marginal probability distribution that must sum to one; please clarify that C is chosen to ensure $\sum_Y p_m(Y)=1$.
- Consider evaluating on larger-scale and more recent datasets, such as DomainNet and Office-Home, to further establish robustness and scalability.

**Questions:**

* What is the upper bound of the method's error, assuming that there is still a certain difference between $P_M(X)$ and $P_T(X)$?
* Can more experiments be provided like semi-synthetic data or larger-scale experiments?

---

### Official Review · Reviewer_yBZx · 2025-11-01

**Soundness:** 2
**Presentation:** 2
**Contribution:** 2
**Rating:** 2
**Confidence:** 5

**Summary:**

The paper proposes a new framework for unsupervised domain adaptation under factorizable joint shift—a setting where both label and covariate distributions change independently. The authors introduce the Label-Covariate Shift Chain (LCSC) representation, decomposing the shift into a sequence of label shift followed by covariate shift, and propose the Covariate Shift Minimization Principle (CSMP) to find non-trivial solutions. The paper also introduces a method for generating synthetic datasets that simulate factorizable joint shifts.

**Strengths:**

Benchmark contribution: The proposed dataset generation pipeline could be useful for future research if properly validated.

Comprehensive experiments: The paper evaluates across multiple datasets and compares with a wide range of baselines, demonstrating empirical thoroughness.

**Weaknesses:**

Theoretical weakness and questionable novelty:
1. The “Label-Covariate Shift Chain” decomposition appears mathematically trivial given the definition of factorizable joint shift (the factorization already implies separability). The theoretical contribution mainly restates this relationship rather than providing fundamentally new insight or provable guarantees.
2. The Covariate Shift Minimization Principle is described heuristically without rigorous justification for why minimizing p_m(X) toward p_t(X) ensures a non-trivial or unique solution.

Methodological ambiguity:
1. Several key quantities (e.g., the construction of p_m(X), choice of \lambda, or estimation of densities via normalizing flows) lack sufficient theoretical grounding or stability analysis.
2. The dependence on accurate high-dimensional density estimation (via normalizing flows) is computationally demanding and potentially unreliable, particularly for large-scale datasets.

Empirical evaluation concerns:
1. The synthetic dataset generation (label + covariate shifts applied sequentially) might not capture realistic or challenging factorizable joint shifts.
2. Improvements in performance are moderate and could plausibly stem from better fine-tuning rather than the proposed theoretical framework.
3. No ablation convincingly isolates the contribution of the Covariate Shift Minimization step versus standard reweighting or resampling baselines.

Positioning and clarity:
1. The connection to prior work (He et al., 2022; Tasche, 2022) is largely incremental.
2. Some theoretical statements (e.g., Theorem 1) are tautological and lack novelty beyond restating known density-ratio decompositions.

**Questions:**

How does LCSC differ substantively from existing density-ratio decomposition methods for joint shifts (e.g., Joint Importance Aligning) beyond reformulation?

How is the “Covariate Shift Minimization Principle” theoretically justified? Is it equivalent to minimizing KL divergence between p_m(X) and p_t(X)?

How sensitive are results to the choice of \lambda, architecture of normalizing flow, or inaccuracies in density estimation?

Could the method be applied when p_t(X) and p_s(X) have disjoint support, or does it require overlap?

How realistic are the generated “factorizable joint shift” datasets compared to genuine domain shifts in the wild?

---

### Official Review · Reviewer_GhYc · 2025-11-01

**Soundness:** 2
**Presentation:** 3
**Contribution:** 2
**Rating:** 4
**Confidence:** 3

**Summary:**

This paper proposes a method to train models to perform well under "factorizable joint shift". This class of shifts assumes that the shift in the joint distribution can be factored into two components that are a function of X and Y exclusively i.e. u(X)v(Y). While this assumption may be restrictive in many scenarios, in generalized pure label or covariate shift. The solution to factorizable joint shift has no unique solution however and the paper proposes "Covariate Shift Minimization Principle" to choose a unique solution amongst the class of possible solutions. Finally, the paper performs some experiments and shows they beat a number of baselines.

**Strengths:**

1. The paper is generally well written
2. The proposed method empirically outperforms a number of baselines
3. The paper's method is derived from a clearly stated principle that makes intuitive sens

**Weaknesses:**

1. I would have liked to see more realistic datasets eg. WILDS or RLSBench. I understand that the assumptions of those tasks may not apply to this method but that is precisely why I would like to see how this method performs on those tasks.

2. The covariate shift seems to be simulated by simple transforms like rotation, translation etc. Would training with data augmentation not make the model robust to this? I would like to see what the results would be of data augmentation + some label shift method like BBSE.

Minor -

3. Some missing references  - [1]

4. Some work has looked at joint shift in terms of latent confounder shift. In the related work, there should be some discussion on whether you can capture such shifts and how you differ from the confounder shift literature - [2][3]

[1]Garg, Saurabh, et al. "Rlsbench: Domain adaptation under relaxed label shift." International Conference on Machine Learning. PMLR, 2023.

[2] Alabdulmohsin, Ibrahim, et al. "Adapting to latent subgroup shifts via concepts and proxies." International Conference on Artificial Intelligence and Statistics. PMLR, 2023.

[3] Prashant, Parjanya Prajakta, et al. "Scalable Out-of-Distribution Robustness in the Presence of Unobserved Confounders." International Conference on Artificial Intelligence and Statistics. PMLR, 2025.

**Questions:**

See Weaknesses

---

### Public Comment · ~Dirk_Tasche1 · 2025-11-13
**Some questions on the paper "Label-covariate shift chain: ..."**

Thanks a lot to the authors for submitting this insightful paper. Still it might be helpful for the readers to see answers to the following questions.

Why is there no "U" term (corresponding to the u(X) part of the u(X) v(Y) factorization) in the optimization objectives (5) and (8) respectively? Without U, the LCSC method looks like just another variant of label shift adaptation.

Is there a need to prove the uniqueness of u(X) v(Y) in Corollary 1? By assumption, u(X) v(Y) is a density of the target distribution with respect to the source distribution. As such u(X) v(Y) is uniquely determined except for events with probability 0 under the source distribution.

Can the authors provide details of how and by whom the (other than LCSC) methods mentioned in Section 5.2.1 were implemented?

---

### Note · Authors · 2025-11-20

I have read and agree with the venue's withdrawal policy on behalf of myself and my co-authors.